**METHOD**

# Simplitigs as an efficient and scalable representation of de Bruijn graphs

Karel Břinda[1,2]* , Michael Baym[1†] and Gregory Kucherov[3,4†]

* Correspondence: karel.brinda@hms.harvard.edu

[†]Michael Baym and Gregory Kucherov contributed equally to this work.

[1]Department of Biomedical Informatics and Laboratory of Systems Pharmacology, Harvard Medical School, Boston, USA and Broad Institute of MIT and Harvard, Cambridge, USA

[2]Center for Communicable Disease Dynamics, Department of Epidemiology, Harvard T.H. Chan School of Public Health, Boston, USA

Full list of author information is available at the end of the article

## Abstract

de Bruijn graphs play an essential role in bioinformatics, yet they lack a universal scalable representation. Here, we introduce simplitigs as a compact, efficient, and scalable representation, and ProphAsm, a fast algorithm for their computation. For the example of assemblies of model organisms and two bacterial pan-genomes, we compare simplitigs to unitigs, the best existing representation, and demonstrate that simplitigs provide a substantial improvement in the cumulative sequence length and their number. When combined with the commonly used Burrows-Wheeler Transform index, simplitigs reduce memory, and index loading and query times, as demonstrated with large-scale examples of GenBank bacterial pan-genomes.

**Keywords:** Sequence analysis, de Bruijn graphs, Simplitigs, *k*-mers, de Bruijn graph representation, Scalability, Pan-genomes, Storage, Data compression, Indexing

## Background

DNA sequencing allowed previously unobservable phenomena to be studied on an unprecedented scale. However, sequencing capacity has grown faster than computer performance, memory, and available human resources, and huge amounts of sequence data are now available. As a result, traditional sequence-based representations and sequence alignment-based techniques [1–3] have become less suitable for real-life scenarios due to the space and time complexities they impose and their inefficiency in handling polymorphisms.

One elegant solution for genomic data representation is de Bruijn graphs. These build on the concept of *k*-mers, which are substrings of a fixed length *k* of the genomic strings to be represented, such as sequencing reads, genomes, and transcriptomes. For a given *k*-mer set, the corresponding de Bruijn graph is a directed graph with the *k*-mers being vertices and *k* – 1 long overlaps between pairs of these *k*-mers indicating edges. If *k* is chosen appropriately, de Bruijn graphs capture substantial information about the sequenced molecules as these correspond to some walks in the graph.

The use of de Bruijn graphs is ubiquitous in sequence analysis. Genome assembly uses the property that sequenced molecules form paths [4–6], which is exploited in numerous modern assemblers [7–12]. On the other hand, alignment-free sequence

comparison follows the idea that similar sequences share common $k$-mers, and comparing de Bruijn graphs thus provides a good measure of sequence similarity [13, 14]. This involves applications of de Bruijn graphs to variant calling and genotyping [15–19], transcript abundance estimation [20], and metagenomic classification [21–24]. In the latter application, $k$-mer-based classifiers perform best among all classifiers in inferring abundance profiles [25], which suggests that de Bruijn graphs truthfully approximate the graph structures of bacterial pan-genomes, even if constructed from noisy assemblies from incomplete databases. Even if more advanced pan-genome graph representations are available, such as variation graphs [26], de Bruijn graphs with large $k$-mer lengths are still useful for indexing [27, 28].

The efficiency of many bioinformatics algorithms is directly tied to the efficiency of computation and representation of the graph. de Bruijn graphs can be readily computed through a scan of the datasets including the raw reads, genomes, or multiple sequence files. In practice, such a scan often consists in $k$-mer counting as this allows efficient denoising of the graph, for example, by removing low-frequency $k$-mers corresponding to sequencing errors in the reads. Algorithms for $k$-mer counting have been extensively studied and many well-engineered software solutions are available [29–37].

On the other hand, efficient representation of de Bruijn graphs remains an important issue. The most commonly used are unitigs, which are strings resulting from compaction of $k$-mers along maximal paths with non-branching nodes [38, 39]. Unitigs have many advantages, including that the representation is "textual," in the form of a set of sequences that contain each $k$-mer exactly once, while preserving graph topology. However, unitigs impose large resource overhead for many types of de Bruijn graphs and do not scale well when a lot of variation is included. Specifically, with a high proportion of branching nodes, unitigs become fragmented, in extreme cases up to the level of individual $k$-mers. Subsequently, unitig computation and storage may require inappropriately large resources and become prohibitive in variation-heavy applications, including bacterial pan-genomics.

While preserving topology guarantees that unitigs represent fragments of the underlying genomic sequences, this property is not required in many bioinformatics applications. Examples are provided by de Bruijn graph storage and data structures for $k$-mer membership queries [38, 40–43], where any set of strings containing the same $k$-mers can be readily used instead. Therefore, the requirements on unitigs could be relaxed in order to create a more lightweight representation carrying the same $k$-mer set but requiring less resources to compute the representation, including CPU time and memory, and having better scalability.

In this paper, we propose simplitigs as a compact, efficient, and scalable representation of de Bruijn graphs. Simplitigs correspond to vertex-disjoint paths covering the graph but relax the unitigs' restriction of stopping at branching nodes. We present an algorithm for rapid simplitig computation from a $k$-mer set and implement it in a tool called ProphAsm, which proceeds by loading a $k$-mer set into memory and a greedy enumeration of maximal vertex-disjoint paths in the associated de Bruijn graph. We used ProphAsm to evaluate the improvement of simplitigs over unitigs, in terms of two main characteristics: the cumulative sequence length (CL) and the number of sequences (NS). We demonstrate that greedily computed simplitigs are close to theoretical bounds in practical applications and, compared to unitigs, provide a substantial improvement in memory requirements and speed in applications such as $k$-mer matching.

## Results

### The concept of simplitigs

We developed the concept of simplitigs to efficiently represent de Bruijn graphs of sequence data (Fig. 1). Simplitigs are a generalization of unitigs and correspond to spellings of vertex-disjoint paths covering a given de Bruijn graph (Fig. 1a, the "Methods" section). Consequently, maximal simplitigs are such simplitigs where no two simplitigs can be merged by a $(k − 1)$ long overlap (the "Methods" section). Note that unitigs and $k$-mers are also simplitigs, but not maximal in general (Fig. 1b). The main conceptual difference between maximal simplitigs and maximal unitigs is that simplitigs are not limited by branching nodes, which allows for further compaction, with a benefit increasing proportionally to the number of branching nodes in the graph.

To compare simplitig and unitig representations, we created a benchmarking procedure based on the two characteristics: the number of sequences (NS) and their cumulative length (CL) (the "Methods" section, example in Fig. 1a). While NS determines the number of records to be kept in memory, CL largely determines the total memory needed. NS and CL are readily bounded from below by one and the number of $k$-mers,

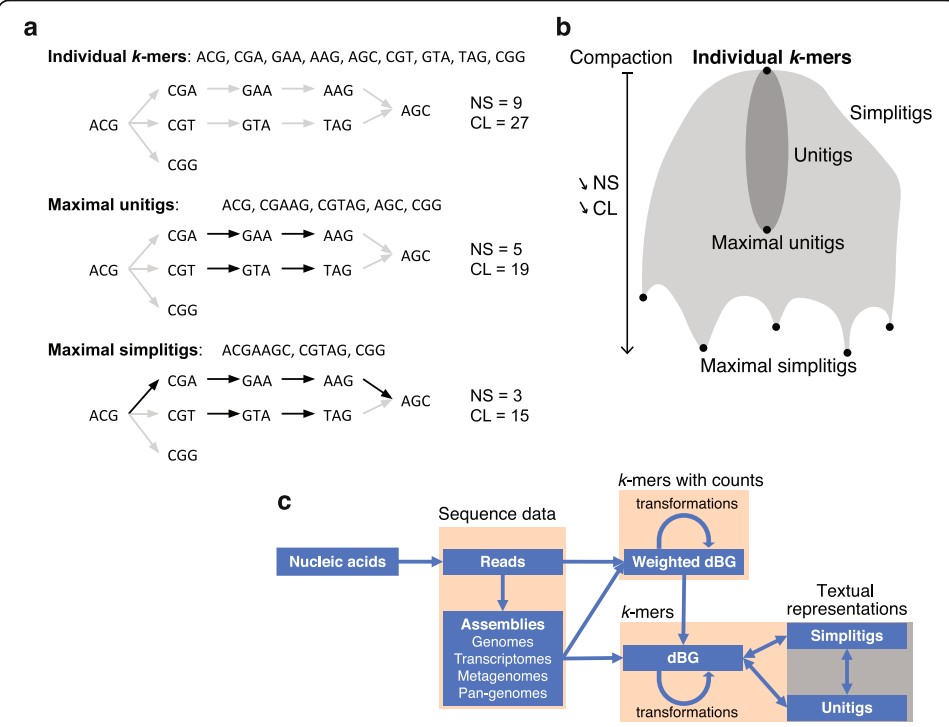

**Fig. 1** Overview of the simplitig approach. **a** Textual representations of $k$-mer sets ordered by the degree of compaction: individual $k$-mers, maximal unitigs, and maximal simplitigs. Every component of a simplitig subgraph (black arrows) of the de Bruijn graph (all arrows) corresponds to a path, and its spelling constitutes a simplitig (the "Methods" section). **b** Scheme of all possible simplitig representations according to the degree of compaction. While unitigs (dark gray area) correspond to compaction along non-branching nodes in the associated de Bruijn graph, simplitigs (gray area) can also contain branching nodes. Every step of compaction decreases the number of sequences (NS) and their cumulative length (CL) by 1 and by $k − 1$, respectively. Maximal simplitigs may not be determined uniquely; the simplitig representation can have multiple local optima, depending on which edges were selected at the branching nodes. **c** The workflow of simplitigs. Simplitigs represent de Bruijn graphs and carry implicitly the same information as unitigs. de Bruijn graphs are usually computed from either assemblies or weighted de Bruijn graphs. Weighted de Bruijn graphs are typically obtained by $k$-mer counting and allow removing noise, e.g., low-frequency $k$-mers, which frequently originate in sequencing errors

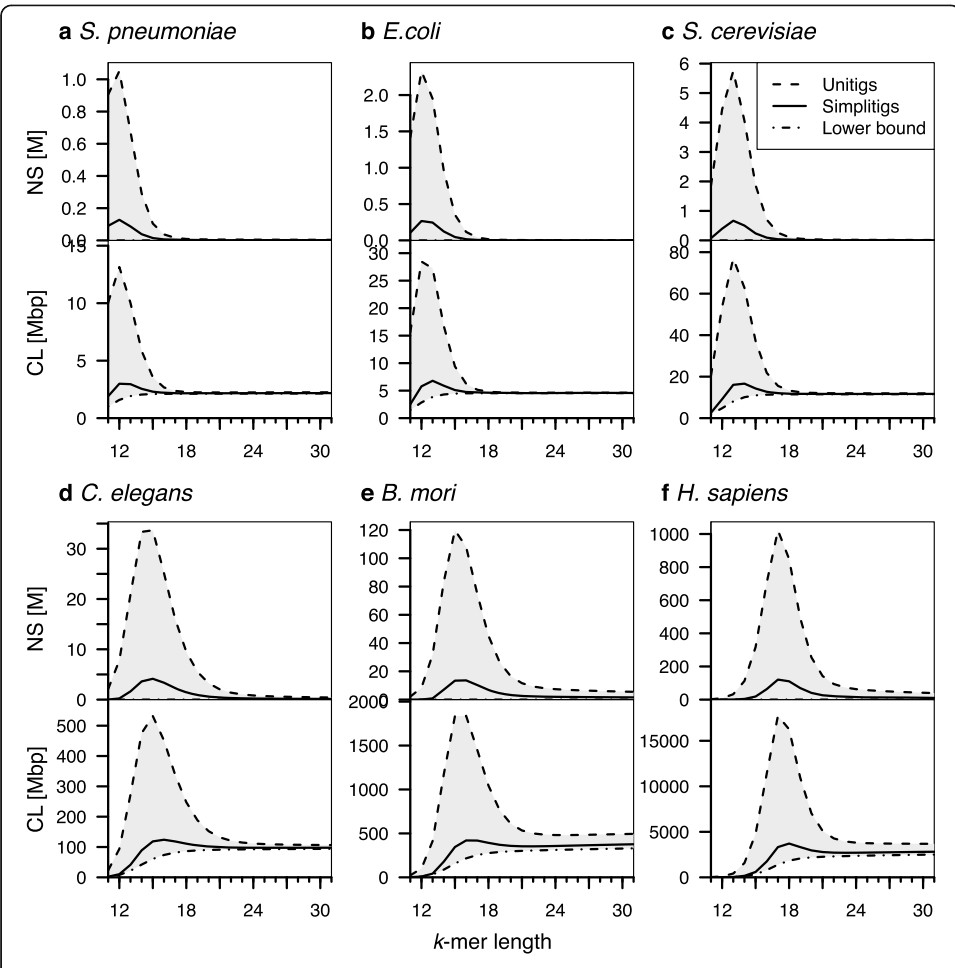

**Fig. 2** Comparison of the simplitig and unitig representations for selected model organisms and a range of *k*-mers. The number of sequences (NS, millions) and their cumulative length (CL, megabase pairs) for both representations of six model organisms ordered by their genome size. **a** *Streptococcus pneumoniae*, 2.22 Mbp. **b** *Escherichia coli*, 4.64 Mbp. **c** *Saccharomyces cerevisiae*, 12.2 Mbp. **d** *Caenorhabditis elegans*, 100 Mbp. **e** *Bombyx mori*, 482 Mbp. **f** *Homo sapiens*, 3.21 Gbp. The CL lower bound corresponds to the number of *k*-mers. Full results are available in Additional file 1: Table S1–S6

respectively, and they are also tightly connected (Eq. 1 in the "Methods" section). As every step of compaction decreases both NS and CL (Fig. 1b, the "Methods" section), we can optimize them jointly. However, finding an optimal simplitig representation translates to the vertex-disjoint path coverage problem. While NP-hard for general graphs (by reduction from the well-known NP-hard problem of computing a Hamiltonian path), the problem may be tractable for observed de Bruijn graphs (the "Methods" section).

Since practical applications do not require optimal simplitigs, we prioritized speed and designed a greedy algorithm for their rapid computation (Algorithm 1, the "Methods" section). In an iterative fashion, the algorithm selects an arbitrary *k*-mer as a seed of a new simplitig and keeps extending it forwards and then backwards as long as possible, while removing the already used *k*-mers from the set; the extension proceeds by all four possible nucleotides and testing for the presence of the created *k*-mer. This process is repeated until all *k*-mers are covered. Loading *k*-mers into memory and simplitig computation are linear in the length of the input and the number of *k*-mers,

respectively, and the memory footprint is linear in the number of $k$-mers. We implemented Algorithm 1 in a program called ProphAsm [44, 45].

---

**Algorithm 1** Greedy computation of maximal simplitigs for a $k$-mer set or a de Bruijn graph. In an iterative fashion, the algorithm draws a $k$-mer from the set of canonical $k$-mers $K$, uses it as a seed of a new simplitig, and then keeps extending the simplitig forwards as long as possible, and then backwards, while removing the already used canonical $k$-mers from $K$.

**Function** extend_simplitig_forwards (K, simplitig):

 extending = **True**

 **while** extending:

 extending = **False**

 q = suffix (simplitig, k**-**1),

 **for** x **in** ['A', 'C', 'G', 'T']:

 canon_kmer = canonical (q + x)

 **if** canon_kmer **in** K:

 extending = **True**

 simplitig = simplitig + x

 K.remove (canon_kmer)

 **break**

 **return** K, simplitig

**Function** compute_maximal_simplitig_from_kmer (K, seeding_kmer):

 simplitig = seeding_kmer

 K, simplitig = extend_simplitig_forwards (K, simplitig)

 simplitig = reverse_completent (simplitig)

 K, simplitig = extend_simplitig_forwards (K, simplitig)

 **return** K, simplitig

**Function** compute_simplitigs (input_kmers):

 K = {}

 **for** kmer **in** input_kmers:

 K.add (canonical (kmer))

 maximal_simplitigs = {}

 **while** |K| > 0:

 seeding_kmer = K.pop ()

 K, simplitig = compute_maximal_simplitig_from_kmer (K, seeding_kmer)

 maximal_simplitigs.add (simplitig)

 **return** maximal_simplitigs

---

## Simplitigs of model organisms

We first evaluated simplitig and unitig representations on assemblies of six model organisms (Fig. 2). As different applications of de Bruijn graphs call for different $k$-mer

lengths, we sought to characterize the NS and CL scaling for both representations with $k$ growing, as well as the effect of the species' genome size. Therefore, selected model organisms were evaluated in increasing order of the genome size and benchmarked for both representations on a range of $k$-mer lengths corresponding to common alignment-free-based applications [20, 21, 46].

We observe that simplitigs always provide a substantially better performance than unitigs (Fig. 2). In particular, they quickly approach the theoretical lower bounds for both characteristics tested. Every data set has a range of $k$-mer lengths where the difference between simplitigs and unitigs is very large, and after a certain threshold, the difference almost vanishes. While for short genomes, this threshold is located at smaller $k$-mer lengths than those typically used in alignment-free applications (e.g., $k \approx 17$ for *E. coli*), for bigger genomes, this threshold has not been attained on the tested range and seems to be substantially shifted towards large $k$-mers (e.g., *B. mori*).

Interestingly, maxima of the NS and CL values for both representations occur very close to the value $k = log_4G$, where $G$ is the genome size (Fig. 2). This is readily explained by edge saturation: for values of $k$ up to $log_4G$, an overwhelming fraction of all $4^k$ $k$-mers belong to the genome, which makes the de Bruijn graph branch at nearly every node. As a consequence, unitigs are then essentially reduced to individual $k$-mers and their number grows exponentially whereas simplitigs stay compact on the whole range of $k$-mer lengths. Starting from $k = log_4G$, the graph starts to form longer non-branching paths, which drives down the NS and CL of unitigs, and they approach those of simplitigs. However, the difference between simplitigs and unitigs in their count and length may stay considerable even for larger values of $k$, especially in case of large eukaryotic genomes.

### Performance assessment

As de Bruijn graph computation can present a bottleneck in bioinformatics pipelines, we compared resources required for computing both representations: unitigs by BCALM and simplitigs by ProphAsm (Fig. 3). We focused on CPU time and memory consumption of both programs when run with 1 thread (the "Methods" section), and we also assessed the effect of parallelization for BCALM by including additional BCALM runs with 4 threads.

On the first five genomes, ProphAsm computation of simplitigs outperformed BCALM computation of unitigs across all $k$-mer lengths and in both CPU time and memory (Fig. 3). For instance, for $k = 31$, ProphAsm was faster by a factor of 15–18, and its memory consumption smaller by a factor of 1.2–2.1. On the genome of *H. sapiens*, ProphAsm was still 10 times faster than BCALM, but its memory footprint was larger by a factor of 1.8, indicative of BCALM's better memory management for large de Bruijn graphs. Interestingly, the memory consumption of BCALM largely improved when we increased the number of threads to four, reflective of different $k$-mer partitioning strategies used with different numbers of threads. We also observed that BCALM resources varied across versions; for instance, version 2.2.3, released after our experiments had been conducted, improved CPU time by a factor 3.4–4.5 for $k = 31$; yet still performing several times slower than ProphAsm (Additional file 2: Table S8). Overall, these results suggest that simplitigs are substantially easier to compute than unitigs.

The resource usage of ProphAsm was also predictable and consistent across experiments; ProphAsm always used 38–51 bytes of RAM per distinct $k$-mer, a limited CPU

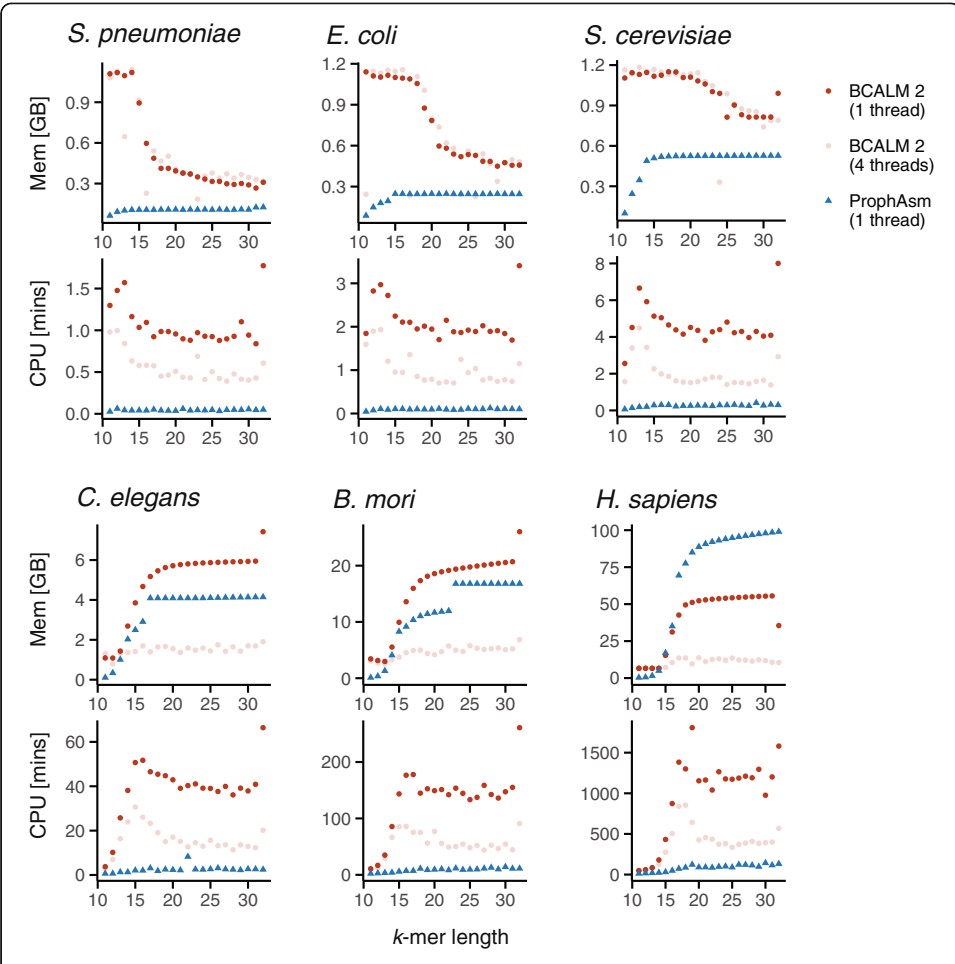

**Fig. 3** Comparison of CPU time and memory consumption of ProphAsm and BCALM. Resources to compute unitigs using BCALM (using one and four threads) and simplitigs using ProphAsm (using one thread) of the six model organisms. Full results are available in Additional file 2: Table S7

time, and no additional disk space. In contrast, BCALM resources for unitigs showed irregular tradeoffs and were difficult to estimate prior to the execution. Therefore, frequent trial-and-error adjustments of the memory and CPU allocations were necessary to even finish some BCALM experiments (the "Methods" section). Furthermore, its high disk space consumption required us to use a cluster partition of large capacity; for instance, a single BCALM run could require as much as 116 GB of disk space (e.g., *H. sapiens*, $k = 17$, and 4 threads). Overall, the experiments suggest that resources for computing unitigs are determined by both the graph size and the complexity of its topology, whereas resources for simplitigs only depend on the graph size; hence, they are more predictable.

### Simplitigs of bacterial pan-genomes
We then evaluated the impact of additional variation in a de Bruijn graph (Fig. 4). Such variation may originate in polymorphisms, varying gene content in a population of genomes that are represented jointly, in haplotypes of viral quasispecies, or in sequencing errors in case of graphs constructed directly from sequencing reads. In all these cases,

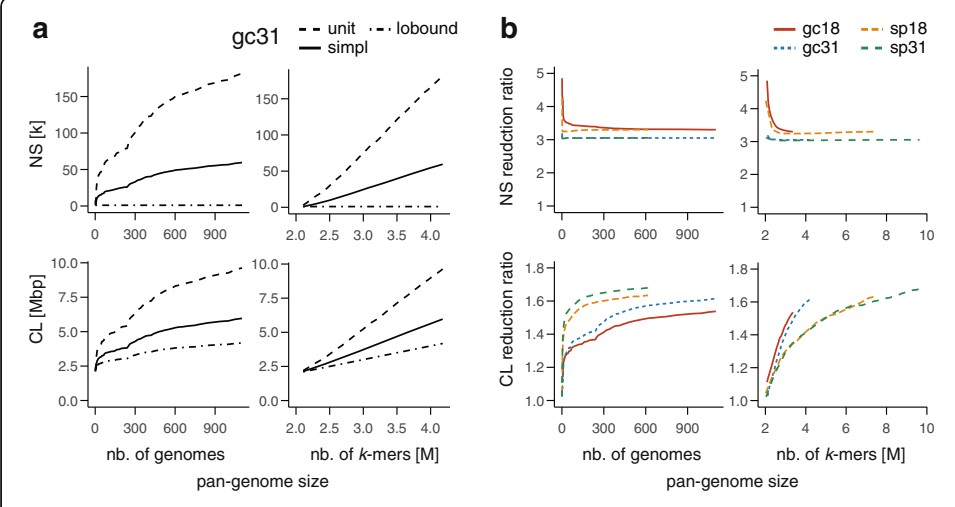

**Fig. 4** Scaling of simplitigs and unitigs of bacterial pan-genomes as the pan-genome size grows with a better sampling and more within-species variation. **a** Number of sequences (NS, thousands) and their cumulative length (CL, megabase pairs) for simplitigs (simpl) and unitigs (unit) and the lower bound (lobound) of *N. gonorrhoeae* and $k = 31$, as a function of the number of genomes (left) and $k$-mers (right, millions) included. **b** Reduction ratio of simplitigs over unitigs for *S. pneumoniae* (sp) and *N. gonorrhoeae* (gc) and $k = 18, 31$, as a function of the number of genomes (left) and $k$-mers (right, millions) included. Full results are available in Additional file 3: Table S11–S14

many nodes of the de Bruijn graph become branching and new paths emerge. To model gradually increasing variation, we used bacterial pan-genomes with different levels of sampling. Given the high diversity and variability of bacteria, de Bruijn graphs provide a convenient option for computational pan-genomes [47]. Such pan-genomes can be constructed from draft assemblies or even directly from sequencing reads, and thanks to bacterial genomes being short and haploid, the information captured by the graphs is sufficient for many analyses.

We first constructed a pan-genome of *N. gonorrhoeae* and characterized unitigs and simplitigs as a function of pan-genome size (Fig. 4a). We used 1102 draft assemblies of clinical isolates from the Gonococcal Isolate Surveillance Project [48], from which we built a series of de Bruijn graphs using an increasing number of genomes. Consistent with previous experiments (Fig. 2a, b, $k = 31$), both representations perform comparably well when only one bacterial genome is included (Fig. 4a). However, as the number of genomes or $k$-mers grows, the NS and CL grow as well, but with an increasing gap between unitigs and simplitigs; importantly, the latter stay close to the theoretical lower bounds. When the pan-genome size is measured via the number of genomes included, the CL and NS resemble logarithmic functions for both unitigs and simplitigs (Fig. 4a, left-hand column). However, when the number of $k$-mers included is used instead, the NS and CL functions act as affine functions (Fig. 4a, right-hand column). This suggests that a pan-genome $k$-mer count and a species-specific slope may be used as the predictors of simplitig performance in future applications.

To analyze the relative benefit of simplitigs with growing de Bruijn graphs, we evaluated the NS and CL reduction ratio of simplitigs over unitigs in different configurations (Fig. 4b). We used the same *N. gonorrhoeae* dataset and considered also another dataset of *S. pneumoniae*, consisting of 616 draft Illumina assemblies of isolates from a carriage study of children in Massachusetts, USA [49, 50]. For both species and for $k = 18, 31$, we constructed a series of de Bruijn graphs as previously, but this time, we visualized

the NS and CL reduction ratios. In all cases, the NS reduction ratios eventually stabilized at values close to 3, following an L shape ($k = 18$) or being almost constant ($k = 31$). The CL reduction ratio admitted approximately a logarithmic dependence on the number of genomes and still resembled a linear dependence on the number of $k$-mers. Overall, these experiments provided further evidence that the benefit of simplitigs over unitigs grows with the increased proportion of branching nodes in a de Bruijn graph or with increasing data in case of pan-genome reference structures.

### Application of simplitigs for de Bruijn graph storage

As simplitigs showed a substantial reduction of CL and NS in genomes and pan-genomes across $k$-mer lengths, we sought to evaluate their applicability for compression (Fig. 5). In this scenario, the goal is to store a given de Bruijn graph within the smallest possible space, measured via the number of bits per distinct $k$-mer (i.e., per node in the graph). Therefore, we reused some of the representations computed in previous experiments for model organisms and bacterial pan-genomes and assessed how these representations can facilitate graph compression. In all cases, we considered two different $k$-mer lengths $k = 18$, 31, for which we computed three text-based representations: simplitigs, unitigs, and assemblies (concatenated in the case of pan-genomes); the obtained sequences were exported to the FASTA format and cleaned to increase their compressibility (the "Methods" section). We also included the BOSS representation [51], as computed using Themisto [52, 53] (the "Methods" section); even though BOSS is a succinct data structure rather than a representation in our sense, its performance can be compared to simplitigs in specific applications. As all of the FASTA and BOSS files contained redundancies, we further compressed them using xz [54], one of the most efficient Unix compressors [55] (the "Methods" section). Finally, we evaluated the compression rate of individual graphs in terms of bits per distinct $k$-mer before and after xz compression, and visualized it on a logarithmic scale (Fig. 5).

We first analyzed the compression rate for the previously studied model organisms (Fig. 5a). In this case, compressed assemblies outperformed the other three approaches, providing near-optimal results. In the case of bacteria (*S. pneumoniae* in Fig. 5a), the three text-based representations provided comparable results for both $k$-mer lengths considered. However, with the genome size growing (see *B. mori* and *H. sapiens* in Fig. 5a), unitigs became increasingly inefficient, especially for $k = 18$, and eventually performed among the worst (consistent with NS and CL in Fig. 2). On the other hand, simplitigs provided a performance comparable to assemblies, with the exception of long genomes or small $k$-mer lengths, in which case assemblies performed better (up to a factor of 2). Even though BOSS initially performed the worst among the four techniques with short genomes, its compression capabilities improved with longer genomes, where it quickly outperformed unitigs and approached simplitigs. Overall, the observed results suggest that if a de Bruijn graph of a single organism is to be compressed, its assembly after cleaning and compression is preferable; on the other hand, if no assembly is available or the downstream applications could suffer from high-frequency $k$-mers, simplitigs present the best solution.

We then performed a similar comparison using the two bacterial pan-genomes (Fig. 5b). In this case, simplitigs always provided the best compression, close to the

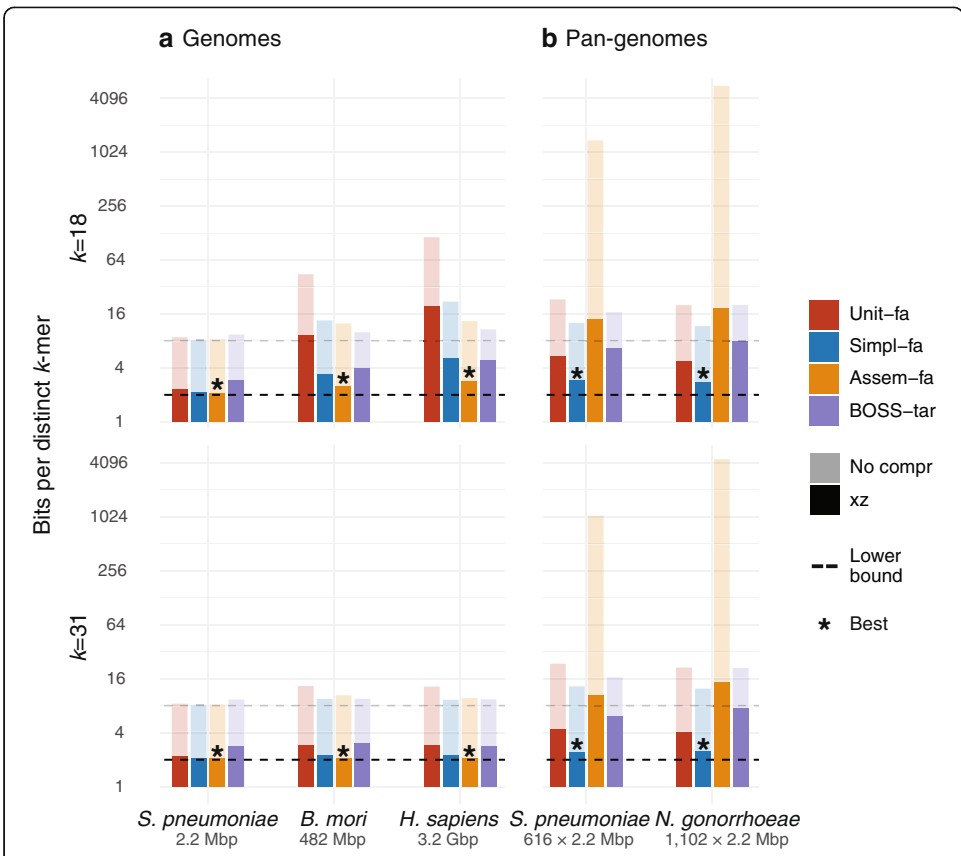

**Fig. 5** Comparison of compression rates of de Bruijn graphs of **a** genomes of model organisms and **b** bacterial pan-genomes, using unitigs, simplitigs, assemblies, and BOSS. The first three representations (text-based) were encoded as cleaned FASTA files, and the BOSS file was obtained as a tar file of all Themisto index files. The compression capabilities of individual representations were compared in terms of the number of bits per distinct $k$-mer for $k = 18$ (top part) and $k = 31$ (bottom part). The results are shown on a logarithmic scale, jointly for uncompressed files (light colors) and files compressed using xz (full colors), together with the lower bounds (8 and 2 bits per $k$-mer for uncompressed and compressed textual representations, respectively). Full results are available in Additional file 4: Table S15.

theoretical minimum of two bits per $k$-mer. We did not observe any significant differences between the two $k$-mer lengths, likely due to the underlying genomes being short (consistent with Fig. 2). Unlike the single-genome experiments, collections of assemblies appeared to be unsuitable for de Bruijn graph compression despite that xz compressed the concatenated assemblies by a factor of 100 and 305, respectively (Additional file 4: Table S16). Furthermore, while BOSS provided comparable results for simplitigs and unitigs in the uncompressed form, its xz compression was much less efficient. Overall, the observed results indicate that for compressing de Bruijn graphs of bacterial pan-genomes, simplitigs are always preferable.

### Application of simplitigs for $k$-mer search

Finally, we sought to demonstrate the benefit of simplitigs in $k$-mer matching, which requires the de Bruijn graph to act as a membership data structure. As both simplitigs and unitigs are text-based representations, $k$-mer queries can be implemented using an arbitrary full-text index [56], notably a Burrows-Wheeler Transform [57] (BWT) index

[58] (sometimes referred to as an FM-index). Here, we used the index of BWA [59], as one of the best-engineered solutions available, to analyze the impact of replacing unitigs by simplitigs.

### Single pan-genome

We evaluated the simplitig improvement on the same *N. gonorrhoeae* pan-genome (Fig. 6). We considered four different *k*-mer lengths *k* = 19, 23, 27, 31, and for each of them, we built the three previously studied pan-genome representations: merged draft assemblies, and simplitigs and unitigs computed for the entire pan-genome. We then used the three representations to construct BWT indexes using BWA [59] and evaluated their performance for *k*-mer matching. As all three indexes carry the same *k*-mer sets, they are equivalent in terms of outputs, but their performance varies due to the different underlying sequences, summarized by NS and CL.

First, we analyzed the NS and CL characteristics of each of the three representations (Fig. 6a). Both simplitigs and unitigs outperformed assemblies in terms of CL by two orders of magnitude; however, while the NS of simplitigs stayed comparable to the NS of assemblies, it increased twofold for unitigs. Compared to unitigs, simplitigs provided an improvement by a factor of 3.1–3.2 and 1.5–1.6 for NS and CL, respectively (consistent with Fig. 4). As the CL characteristic corresponds to the size of the composite sequence in the index, we estimated that the simplitig indexes should consume 34–38% less memory. However, the measured values showed a different picture—switching from unitigs to simplitigs decreased memory consumption by 64–67% (Fig. 6b, top part). This suggests that not only the total length, but also the number of sequences largely determines the performance of a BWT index; therefore, the CL and NS characteristics should always be studied jointly when text-based de Bruijn graph representations are being compared. We then evaluated time to match 10 million *k*-mers using the BWA fastmap command [60] and found that matching with simplitigs was faster by

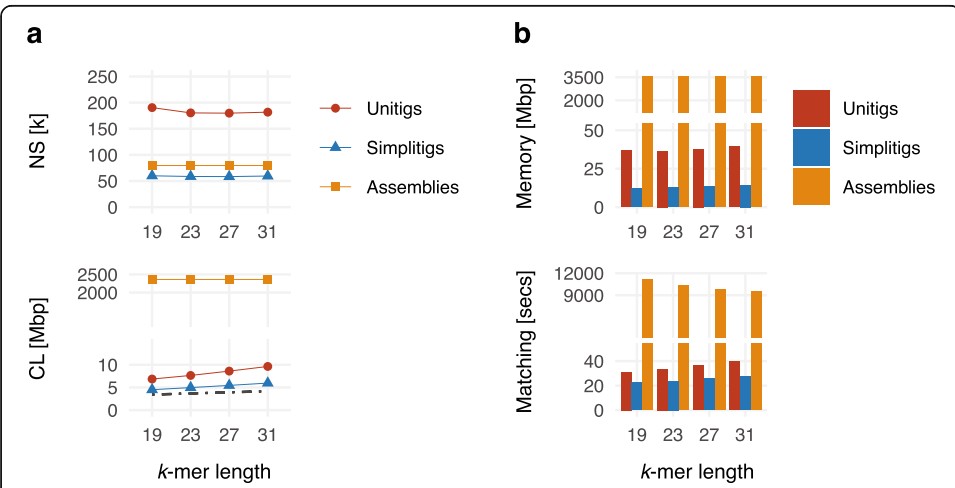

**Fig. 6** *k*-mer queries for the *N. gonorrhoeae* pan-genome on top of the draft assemblies, unitigs, and simplitigs. **a** Characteristics of the obtained unitigs and simplitigs: number of sequences (NS, thousands) and their cumulative length (CL, megabase pairs). The dot-dash line depicts the CL lower bound corresponding to the number of *k*-mers. **b** Memory footprint and time to match 10 million *k*-mers using BWA. Full results including relative improvements are available in Additional file 5: Table S17–S18

a factor of 1.38–1.45 (Fig. 6b, bottom part). The observed speedup likely arose from a combination of factors, including the lower number of sequences (NS), the reduction of high-frequency spurious $k$-mers on sequence borders in the composite sequence, and also the lower memory footprint. Overall, the experiment showed that simplitigs substantially improve the performance in $k$-mer matching compared to unitigs in terms of both memory and speed.

### Multiple pan-genomes

We also evaluated the performance of the simplitig representation for simultaneous indexing of a large number of bacterial pan-genomes (Fig. 7). We downloaded all complete bacterial genomes from GenBank that had not been excluded from RefSeq (as of May 2020; 9869 records out of which 9032 had genomic sequences available; the "Methods" section). We restricted ourselves to complete genomes as draft genomes in GenBank are substantially impacted by false genetic variability [61–64]. By grouping individual genomes per species, we obtained 3179 bacterial pan-genomes which we call the "All" dataset. After computing simplitigs and unitigs per species, we merged the obtained representations and constructed indexes using BWA; all this was done for $k = 19, 23, 27, 31$ to evaluate the impact of the $k$-mer length. As none of the unitig indexes could fit into the RAM of our desktop computer, we also created the "Solid" dataset by omitting pan-genomes with less than 11 genomes; this resulted in 112 pan-genomes with 3958 genomes. We provide all the constructed pan-genomes in the form of simplitigs on Zenodo [65].

First, we analyzed the obtained simplitig and unitig representations of both datasets (Fig. 7a). We observe that simplitigs provided a substantial improvement in both test characteristics. In the Solid dataset, NS and CL were reduced by simplitigs by a factor of 3.1–4.5 and 1.4–1.9, respectively, and in the All dataset, NS and CL were reduced by a factor of 3.0–4.3 and 1.2–1.4, respectively, consistently with the scaling observed previously (Fig. 4, Fig. 6). While the improvement in NS was almost identical in both datasets (consistent with the top-right graph in Fig. 4b), the improvement in CL was clearly better in the Solid dataset. Indeed, as the vast majority of pan-genomes in the All dataset contained only one genome, the de Bruijn graphs had a comparatively low number of branching nodes; therefore, the difference between simplitigs and unitigs was less striking (consistent with the values for small pan-genome sizes in Fig. 4b). We also observe that, in contrast to unitigs, $k$-mer length had only little impact on the CL of simplitigs within the tested range, which provides better guarantees on required computational resources in future applications.

We then measured the performance of $k$-mer lookup (Fig. 7b). Both on a desktop and on a cluster, we evaluated memory footprints, index loading time, and time to match ten million random $k$-mers from the index using BWA (the "Methods" section). We observed that simplitigs substantially improved the memory footprint and index loading times. For $k = 19$, simplitigs largely improved the matching times, where the difference was caused by spurious $k$-mers on unitig borders; these were more common in this experiment due to the short $k$-mer length and the high number of unitigs. For higher $k$-mer lengths, simplitigs still provided a moderate improvement in the matching rate. We note that the query time with BWT-based $k$-mer indexes is dominated by

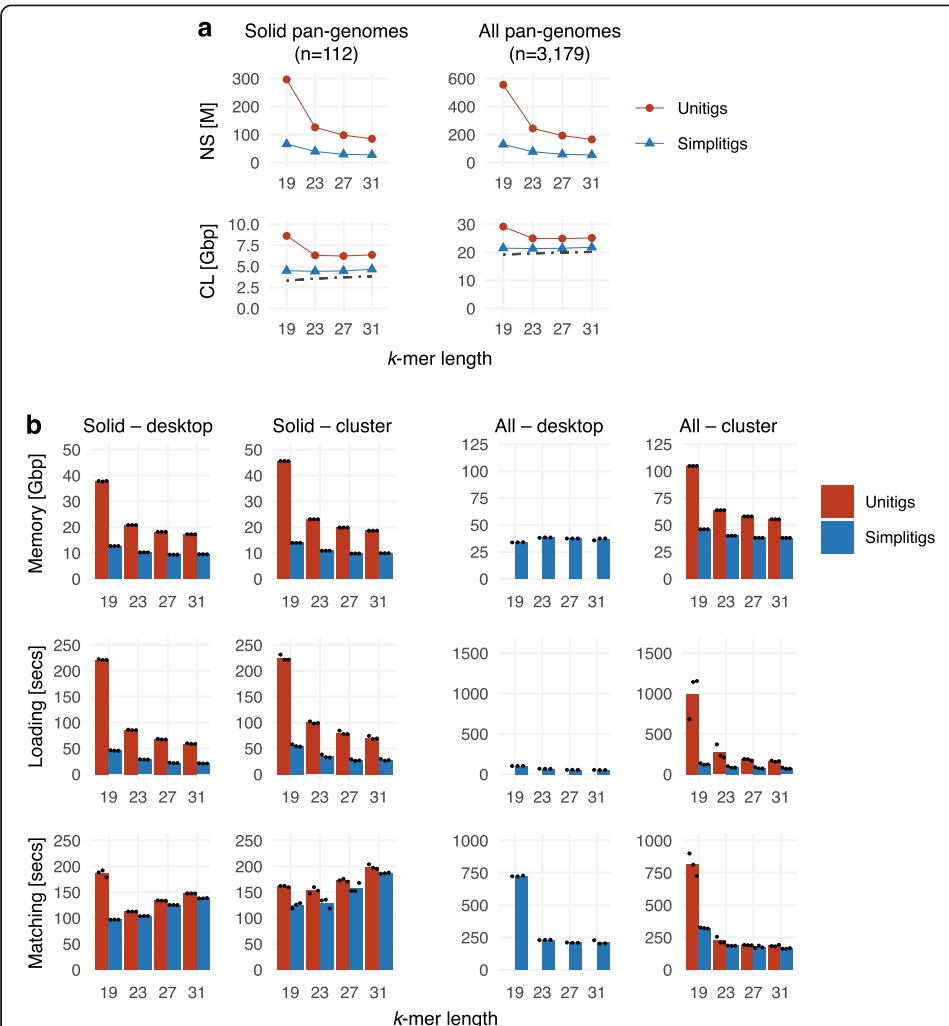

**Fig. 7** *k*-mer queries for multiple pan-genomes indexed simultaneously. Bacterial pan-genomes were computed from the complete GenBank assemblies per individual species. While the All dataset comprises all pan-genomes with no restriction on their size, the Solid dataset comprises only those that contain at least 11 genomes. **a** Characteristics of the obtained unitigs and simplitigs: number of sequences (NS, millions) and their cumulative length (CL, gigabase pairs). The dot-dash line depicts the lower bound corresponding to the number of *k*-mers. **b** Memory footprint, time of index loading, and time of matching 10 million *k*-mers using BWA. The bars correspond to the mean of three measurements (black dots). Full results including relative improvements are available in Additional file 6: Table S19–S23

high-frequency *k*-mers; the observed performance differences are thus likely to be a consequence of the spurious *k*-mers on sequence borders as seen previously. On the desktop, the unitig All-index could not be evaluated as it did not fit into memory, and the outlier for simplitigs with $k = 19$ may be the result of memory swapping.

## Discussion

We introduced the concept of simplitigs, a light-weight alternative to unitigs, and demonstrated that simplitigs constitute a compact, efficient, and scalable representation of de Bruijn graphs for various types of genomic datasets. The two representations share many similarities: they are text-based, and individual strings correspond to spellings of vertex-disjoint paths. Both representations can be seen as irreversible transforms, taking

a set of input strings and producing a new set of strings preserving the *k*-mer sets. In both cases, the resulting files can easily be manipulated using common Unix tools, indexed using full-text indexes, and further compressed using standard compression techniques. The main difference is that simplitigs, unlike unitigs, do not explicitly carry information about the topology of the de Bruijn graph. Furthermore, simplitigs are not expected to have direct biological significance—adjacent segments of the same simplitig may correspond to distant parts of the same nucleic acid or even to different ones. Nevertheless, unitigs can always be recomputed from simplitigs, but this step is not required for many common applications. Moreover, a concept analogous to simplitigs, called disjointigs, was recently introduced in the context of genome assembly using A-Bruijn graphs [66, 67], suggesting that simplitigs may be useful beyond the context of topology-oblivious applications.

We efficiently computed maximal simplitigs from *k*-mer sets using ProphAsm, a tool implementing a greedy heuristic. ProphAsm is a spin-off of the ProPhyle software (https://prophyle.github.io, [23, 68]) for metagenomic classification, allowing efficient indexing of *k*-mers propagated to individual nodes of a phylogenetic tree. ProphAsm presents a "naive" implementation of the greedy heuristic (Algorithm 1) that can be further improved. For instance, a hash table with better memory management can reduce the memory requirement by a factor of 2.5 [69] and additional memory reduction can be achieved similarly to previous unitig implementations [39, 40, 70]. On the studied data, ProphAsm outcompeted BCALM, the most advanced tool for unitigs, in all characteristics measured, with the sole exception of memory in the case of *H. sapiens*. This suggests, on the one hand, that simplitigs are generally easier to compute, and, on the other hand, that tools driven by assembly-centric applications can become inefficient when graphs contain too many branching nodes, for instance, due to a small *k*-mer length. The achieved simplicity of simplitigs and ProphAsm makes them easily applicable to batch jobs deployed in parallel on a cluster thanks to the ease of resource predictability. However, once the number of *k*-mers in a dataset exceeds a critical threshold corresponding to the maximum RAM available, more sophisticated computational approaches using *k*-mer partitioning will become necessary.

A challenging but also promising feature of simplitig representation is the ambiguity of maximal simplitigs. This is in sharp contrast to maximal unitigs, which are uniquely defined (up to the order, reverse complementing, and cycles). In practice, every algorithm for simplitig computation has to decide which edge will be included at each branching node. Here, we prioritized speed, the simplitigs were constructed progressively, and lexicographically minimal edges were applied in the case of ambiguity. Therefore, final maximal simplitigs were dependent only on the choice of seeding *k*-mers, which are determined by the specific implementation of "std::unordered_set" in the C++ standard library. Nevertheless, characteristics other than speed could readily be prioritized. For example, a more sophisticated heuristic could drive CL and NS closer to the optimum and thus improve compressibility. One could also aim at adding biological significance to simplitigs; for instance, by preferring those paths that are better supported by sequence data. Finally, future streaming algorithms for operations such as merging or intersecting may require specific prescribed forms of simplitigs.

We note that the simplitig representation was independently and simultaneously studied under the name "spectrum-preserving string sets" (SPSS) [71]. Both

representations are conceptually equivalent, corresponding to spelling maximal vertex-disjoint paths in the graph. Similar to ProphAsm, the associated UST software computed simplitigs using a greedy approach on top of a de Bruijn graph. On the other hand, ProphAsm and UST come with different complexities and bottlenecks: whereas ProphAsm computes simplitigs directly from $k$-mer sets, the UST pipeline first computes unitigs using BCALM. However, as we showed in the "Performance assessment" section, BCALM performed substantially worse than ProphAsm on our data in all characteristics considered (except memory for *H. sapiens*). Similar to our approach, [71] compares representations using the cumulative length of the representation (therein termed weight), but the number of sequences (NS) was not considered. On the other hand, [71] established a tighter lower bound on CL compared to ours, taking into account the graph topology but requiring a computational overhead. Another major difference between our work and [71] is in the data used for evaluation: while we used $k$-mer sets from genome assemblies, [71] considered raw reads with frequency-based $k$-mer filtering; as these datasets are fundamentally different, the presented quantitative measures are not directly comparable across the two papers.

Our work opens many questions and future directions. On the theoretical side, we anticipate advances in the analysis of the minimum vertex-disjoint path cover problem, a better connection to results from other disciplines such as network sciences, and improvements in the resulting heuristic that could also facilitate parallelization and streaming. The nature of the algorithm implies that simplitigs can be computed online, directly from a stream of data such as sequencing reads. Promising directions are related to different algorithmic techniques, such as sketching [72, 73], and other string processing concepts, such as minimal absent words [74], shortest superstrings [75], and textual transforms such as the Burrows-Wheeler Transform [57]. On the practical side, we anticipate better implementations of simplitig computation, which could also be plugged into standard bioinformatics libraries for various programming languages. Another series of questions is related to low-memory transformations of simplitigs that would allow precomputing simplitigs on computational clusters and tailoring them to specific applications on standard computers; this includes decreasing $k$, performing set operations on top of simplitig sets, and computing maximal unitigs from maximal simplitigs. A substantial body of work can be anticipated in the direction of text indexing—we showed that simplitigs can be combined with full-text indexes; however, specialized indexes fully exploiting simplitig properties are yet to be developed. Finally, indexed simplitigs of bacterial pan-genomes could provide a lightweight alternative to complex tools implementing read mapping and quasi-mapping to reference graphs, such as VG [28] or Pandora [76].

In modern bioinformatics applications, de Bruijn graphs are oftentimes employed as core data structures [77], and simplitigs can be used as their components. For instance, data structures for membership queries [78] relying on unitigs [38, 40–43] could be redesigned to use simplitigs instead. In many applications, including some of the traditional alignment-free methods [13, 14], it is desirable to consider $k$-mers with counts, which leads to so-called weighted de Bruijn graphs [79]; a recent manuscript [80] introduced monotigs which are a form of short simplitigs to encode this information. Furthermore, multiple de Bruijn graphs are often considered simultaneously; the resulting structure is usually referred to as a colored de Bruijn graph [15] and the associated

data structures have been also widely studied [41, 43, 51, 81–89]. Although we touched upon this issue in the "Multiple pan-genomes" section, exploiting the similarity between individual de Bruijn graphs for further compression in simplitig-based approaches will be addressed in our future work.

## Conclusions

We addressed the question of efficient and scalable representation of de Bruijn graphs. We showed that unitigs, the state-of-the-art representation, may require inadequately large computational resources, especially when de Bruijn graphs contain many branching nodes. We introduced simplitigs, which provide a more compact replacement in applications that do not require explicit information on the graph topology, such as alignment-free sequence comparison and $k$-mer indexing. We introduced a heuristic for simplitig computation and showed on the examples of model species that unless the genome is large, even a naive implementation of simplitigs outperforms BCALM, the main state-of-the-art tool for unitigs. We then studied applications to bacterial pan-genomics and showed that the utility of simplitigs compared to unitigs grows as more data are available. We also studied the applications of simplitigs to de Bruijn graph storage and showed that in the case of single genomes, assemblies provide a better compression rate, but for bacterial pan-genomes, simplitigs are preferential. Finally, we demonstrated on the example of full-text $k$-mer indexing that simplitigs can substantially reduce computational resources in downstream applications. Our experiments also highlight a better scaling of simplitigs for growing datasets: as we have shown in the paper, with more data available, the comparative benefits of simplitigs over unitigs become more important. While the growth of public databases negatively impacts the accuracy of algorithms using inexact representations (e.g., those based on LCA) [90], simplitigs provide a promising solution offering both exactness and scalability. Therefore, we envision the simplitig representation and its derivatives to become a generic compact representation of de Bruijn graphs, in particular, in the context of large-scale sequence data search engines [88, 91] and repositories such as those of NCBI and EBI.

## Methods

### de Bruijn graphs

All strings are assumed to be over the alphabet $\{A, C, G, T\}$. A $k$-mer is a string of length $k$. For a string $s = s_1 \cdots s_n$, we define $pref_k(s) = s_1 \cdots s_k$ and $suf_k(s) = s_{n-k+1} \cdots s_n$. For two strings $s$ and $t$ of length at least $k$, we define the binary connectivity relation $s \rightarrow_k t$ if and only if $suf_k(s) = pref_k(t)$. If $s \rightarrow_k t$, we define the $k$-merging operation $\odot^k$ as $s \odot^k t = s \cdot suf_{|t|-k}(t)$.

Given a set $K$ of $k$-mers, the de Bruijn graph of $K$ is the directed graph $G = (V, E)$ with $V = K$ and $E = \{(u, v) \in K^2 \mid u \rightarrow_{k-1} v\}$. For every path $p = (v_1, \ldots, v_p)$ in $G$, the string $v_1 \odot^{k-1} v_2 \odot^{k-1} \ldots \odot^{k-1} v_p$ is called a spelling of $p$. This definition of de Bruijn graphs is node-centric, as nodes are identified with $k$-mers and edges are implicit, and we can use the terms "$k$-mer set" and "de Bruijn graph" interchangeably.

### Simplitigs

Consider a set $K$ of $k$-mers and the corresponding de Bruijn graph $G = (K, E)$. A *simplitig graph* $G' = (K, E')$ is a spanning subgraph of $G$ that is acyclic and the in-degree and out-degree of any node is at most one. It follows from this definition that a simplitig graph is a vertex-disjoint union of paths, whose spellings we call *simplitigs*. A simplitig is called *maximal* if it cannot be extended forward or backward without breaking the definition of simplitig graph. In more detail, a simplitig $u_1 \rightarrow_{k-1} u_2 \rightarrow_{k-1} \ldots \rightarrow_{k-1} u_n$ is maximal if the following conditions hold:

- Either $u_1$ has no incoming edges in $G$, or for any edge $(v, u_1) \in E$, $v$ belongs to another simplitig and it is not its last vertex
- Either $u_n$ has no outgoing edges in $G$, or for any edge $(u_n, v) \in E$, $v$ belongs to another simplitig and it is not its first vertex

A *unitig* is a simplitig $u_1 \rightarrow_{k-1} u_2 \rightarrow_{k-1} \ldots \rightarrow_{k-1} u_n$ such that each of the nodes $u_2$, ..., $u_n$ has in-degree 1 and each of the nodes $u_1$, ..., $u_{n-1}$ has out-degree 1 in graph $G$. A maximal unitig is defined similarly.

### Comparing simplitig and unitig representations

Simplitig and unitig representations were compared in terms of the number of sequences produced (NS) and their cumulative length (CL). For any set of simplitigs (i.e., not necessarily maximal ones), NS is bounded by 1 and #*kmers*, CL is bounded by #*kmers* and $k \cdot$ #*kmers*. The upper bound corresponds to the state of maximal fragmentation, where every $k$-mer forms a simplig. The lower bound corresponds to the maximum possible degree of compaction, i.e., a single simplitig containing all $k$-mers.

NS and CL are readily connected by the following formula:

$$CL = \#kmers + (k - 1) \cdot NS \tag{1}$$

As an important consequence, both characteristics are optimized simultaneously.

### Greedy computation of simplitigs

The problem of computing maximal simplitigs that are optimal in CL (i.e., also in NS) corresponds to the minimum vertex-disjoint path cover problem [92]. This is known to be NP-hard in the general case, by reduction from the Hamiltonian path problem. However, the complexity for de Bruijn graphs remains an open question. A greedy heuristic to compute maximal simplitigs has been used throughout this paper (Algorithm 1). Simplitigs are constructed iteratively, starting from (arbitrary) seeding $k$-mers and being extended greedily forwards and backwards as long as possible.

### ProphAsm implementation

ProphAsm is written in C++ and implements the greedy approach described above (Algorithm 1). $k$-mers are encoded using uint64_t and stored in an std::unordered_set. The choice of extension nucleotides on branching nodes is done based on the lexicographic order. Therefore, the only source of randomness is the choice of seeding $k$-mers by std::unordered_set::begin; the C++ standard library makes no guarantees on which specific element is considered the first element. ProphAsm

does not require any disk space to store intermediate data and its memory consumption corresponded to 38–51 bytes per a unique $k$-mer (in dependence on the allocation), consistent with [69].

### Uni-directed and bi-directed models

The uni-directed model, as presented above, is useful for introducing the concepts of unitigs and simplitigs, but it is not directly applicable to data obtained using sequencing: since DNA is double-stranded, every string may come from either strand. At the level of $k$-mers, double-strandedness can be accounted for by using canonical $k$-mers, i.e., by pairing-up every $k$-mer with its reverse complement, typically done by taking the lexicographical minimum of the $k$-mer and its reverse complement. This subsequently requires redefining de Bruijn graphs to bi-directed de Bruijn graphs [93], which requires a more complex formalism.

### Correctness evaluation

The correctness of simplitig computation can be verified using an arbitrary $k$-mer counter. Simplitigs have been computed correctly if and only if every $k$-mer is present exactly once and the set of distinct $k$-mers is the same as in the original datasets. The correctness of ProphAsm outputs was verified using JellyFish 2 [30].

### Experimental evaluation—model organisms and performance

Reference sequences for six selected model organisms were downloaded from RefSeq and UCSC Genome Browser: *S. pneumoniae* str. ATCC 700669 (accession: NC_011900.1, length 2.22 Mbp), *E. coli* str. K-12 (accession: NC_000913.3, length: 4.64 Mbp), *S. cerevisiae* (accession: NC_001133.9, length: 12.2 Mbp), *C. elegans* (accession: GCF_000002985.6, length: 100 Mbp), *B. mori* (accession: GCF_000151625.1, length: 482 Mbp), and *H. sapiens* (HG38, http://hgdownload.soe.ucsc.edu/goldenPath/hg38/bigZips/hg38.fa.gz, length: 3.21 Gbp). For each genome, simplitigs and unitigs were computed using ProphAsm and BCALM, respectively, for a range of $k$-mer lengths [11..32].

Individual experiments were run in parallel on the Harvard Medical School O2 cluster using Snakemake [94] and SLURM. ProphAsm and BCALM were run with the following parameters, respectively: "-k {kmer-length}" and "-kmer-size {kmer-length} -abundance-min 1 -nb-cores {cores} -max-disk 30000." As BCALM requires a large undocumented amount of disk space, we used the -max-disk parameter to make a parallel execution of many BCALM jobs feasible. The SLURM specifications of resource allocation for individual species were iteratively adjusted until all jobs would finish; the final required resources are provided in Additional file 2: Table S9. Time and memory consumption of jobs were measured independently using GNU Time. Individual jobs were deployed to computational nodes with different hardware configurations, which are specified in Additional file 2: Table S10.

### Experimental evaluation—bacterial pan-genomes

First, 1102 draft assemblies of *N. gonorrhoeae* clinical isolates (collected from 2000 to 2013 by the Centers for Disease Control and Prevention's Gonococcal Isolate Surveillance Project [48], and sequenced using Illumina HiSeq) were downloaded from

Zenodo [95]. Second, 616 draft assemblies of *S. pneumoniae* isolates (collected from 2001 to 2007 for a carriage study of children in Massachusetts, USA [49, 50], and sequenced using Illumina HiSeq) were downloaded from the SRA FTP server using the accession codes provided in Table 1 in [50]. For each of these datasets, an increasing number of genomes were being taken and merged, and simplitigs and unitigs computed using ProphAsm and BCALM, respectively. This experiment was performed for $k = 18$ and $k = 31$. To avoid excessive resource usage the functions were evaluated at selected points in an increasing distance: for intervals [10..96] and [100..+∞] only multiples of 5 and 20 were evaluated, respectively.

### Experimental evaluation—compression

The same six model organisms and two bacterial pan-genomes as above were used for the evaluation of compression. In the case of the pan-genomes, the 616 and 1102 assemblies of *S. pneumoniae* and *N. gonorrhoeae* were merged into single files, respectively. Simplitigs and unitigs were computed using ProphAsm and BCALM, respectively, as previously. All the obtained FASTA files were converted to the same cleaned format: first, sequences were split at unknown nucleotides using Seqtk [96] (v1.3-r106, "seqtk cutN -n1"), sequences shorter than 18 bp discarded, all sequences converted to the one-line format ("seqtk seq -U -C -L18") and renamed to sequentially assigned numbers using bioawk (v1.0–20110810, "bioawk -c fastx" {{ print ">" ++i" \\n"$seq }}"). The BOSS representation [51] was computed using Themisto [52, 53] (commit 21a48ec, "build_index --mem-megas 20000 --k {kmer-length} --input-file {input-file} --n-threads 8") and the obtained files merged using tar. The xz software was applied to the FASTA and BOSS files with the best compression level and a single thread (XZ Utils v5.2.5, "xz -T1 -v -9"). Finally, the sizes of the obtained files were computed using wc ("wc -c").

### Experimental evaluation—full-text *k*-mer queries

In the single pan-genome experiment, the same 1102 assemblies of *N. gonorrhoeae* were merged into a single file. ProphAsm and BCALM were then used to compute simplitigs and unitigs, respectively, from this file for $k = 19, 23, 27, 31$. Each of the three obtained FASTA files (assemblies, simplitigs, and unitigs) was used to construct a BWA index, which was then queried for *k*-mers using "bwa fastmap -l {kmer-length} -w 99999". The "-w" parameter (the maximum size of suffix-array intervals) was necessary for ensuring evaluation correctness; otherwise, certain *k*-mers would be discarded from matching, inconsistently across the three considered representations. We used a modified version of BWA fastmap that reports both the time of index loading and the time of querying (http://github.com/karel-brinda/bwa, commit e1f907c). Query *k*-mers were generated from the same pan-genome using WGsim (version 1.10, "wgsim -h 0 -S 42 -r 0.0 -1 {kmer-length} -N 10000000 -e 0").

For the multiple pan-genome experiment, a list of available bacterial assemblies was downloaded from ftp://ftp.ncbi.nlm.nih.gov/genomes/genbank/bacteria/assembly_summary.txt (2020/05/05). For all assemblies marked as complete (i.e., the "assembly_level" column equal to "Complete genome") and present in RefSeq (i.e., an empty value in the column "excluded_from_refseq"), directory URLs and species names were extracted ($n = 9869$). These were then used to download the genomes of the isolates using

RSync, restricted to genomic sequences only (i.e., files matching "*v?_genomic.fna.gz," $n = 9032$). The downloaded assemblies were then merged per species in order to collect $k$-mers of individual pan-genomes and used for computing simplitigs and unitigs using ProphAsm and BCALM, respectively. The obtained simplitig and unitig files were then merged per categories (e.g., simplitigs for $k = 19$) and used to construct BWA indexes. The obtained indexes were queried for 10 million $k$-mers using BWA fastmap as previously. The $k$-mers were generated from the original assemblies of randomly selected 100 genomes using DWGsim [97] (version 0.1.11, "dwgsim -R 0 -e 0 -r 0 -X 0 -y 0 -H -z 42 -m /dev/null -N 10000000 -1 {kmer-length} -2 0"); the randomization was performed using "sort -R".

### Computational setup

The experiments were performed on the HMS O2 research high-performance cluster and on an iMac 4.2 GHz Quad-Core Intel Core i7 with 40 GB RAM. The reproducibility of computation was ensured using BioConda [98]. All benchmarking was performed using ProphAsm 0.1.1 (commit ea28b708) and BCALM 2.2.2 (commit febf79a3); in the BCALM version experiment, its performance was compared to BCALM 2.2.3 (commit 1f8a8b1). Time and memory footprint were measured using GNU Time.

## Supplementary Information

**Additional file 1.** Detailed information for the single genome experiment: NS, CL and #$k$-mers for unitigs and simplitigs as a function of $k$ for the 6 species. **Supplementary Table S1.** *S. pneumoniae.* **Supplementary Table S2.** *E. coli.* **Supplementary Table S3.** *S. cerevisiae.* **Supplementary Table S4.** *C. elegans.* **Supplementary Table S5.** *B. mori.* **Supplementary Table S6.** *H. sapiens.*

**Additional file 2.** Detailed information for the performance comparison. **Supplementary Table S7.** CPU time and memory consumption (both measured by GNU Time and Snakemake) as a function of species, method, number of threads, and $k$-mer length, including the used computational node. **Supplementary Table S8.** Comparison of BCALM versions (CPU time with v2.2.2 vs. v2.2.3). **Supplementary Table S9.** SLURM resource allocation for ProphAsm and BCALM for the performance evaluation. **Supplementary Table S10.** Hardware specifications for individual computational nodes.

**Additional file 3.** Detailed information for the pan-genome scaling experiment: **Supplementary Table S11.** *N. gonorrhoeae*, $k = 18$ (gc18). **Supplementary Table S12.** *N. gonorrhoeae*, $k = 31$ (gc31). **Supplementary Table S13.** *S. pneumoniae*, $k = 18$ (sp18). **Supplementary Table S14.** *S. pneumoniae*, $k = 31$ (sp31).

**Additional file 4.** Detailed information for the compression experiment. **Supplementary Table S15.** Size characteristics of the stored de Bruijn graphs for single genomes of 6 species (*S. pneumoniae, E. coli, S. cerevisiae, C. elegans, B. mori,* and *H. sapiens*) and 2 bacterial pan-genomes (*S. pneumoniae* and *N. gonorrhoeae*) for each of the 4 studied methods, $k = 18, 31$, before and after compression using xz. **Supplementary Table S16.** The resulting compression ratios for individual experiments.

**Additional file 5.** Detailed information for the single *N. gonorrhoeae* pan-genome $k$-mer indexing experiment. **Supplementary Table S17.** Characteristics of the resulting pan-genome simplitigs and unitigs for $k = 19,23,27,31$. **Supplementary Table S18.** Performance of $k$-mer querying for the pan-genome: memory footprint, index loading time and time to query 10 million $k$-mers using BWA.

**Additional file 6.** Detailed information for the multiple pan-genomes $k$-mer indexing experiment. **Supplementary Table S19.** List of all genomes used for building the pan-genomes (accession code, version, species, filename, number of sequences, genome size [bp]); **Supplementary Table S20.** List of species and the number of genomes included. **Supplementary Table S21.** Characteristics of the resulting simplitigs and unitigs of individual species pan-genomes for $k = 19,23,27,31$. **Supplementary Table S22.** Characteristics of the resulting simplitigs and unitigs for the All-dataset and Solid-dataset and $k = 19,23,27,31$. **Supplementary Table S23.** Performance of $k$-mer querying for the pan-genome: memory footprint, index loading time and time to query 10 million $k$-mers using BWA (individual repetitions).

**Additional file 7.** Review history.

### Acknowledgements

The authors thank Jasmijn Baaijens, Roman Cheplyaka, Donald Halstead, Paul Medvedev, and Rayan Chikhi for their valuable comments and Kamil Salikhov and Simone Pignotti for the helpful discussions at the initial stage of this

project. Portions of this research were conducted on the O2 high-performance compute cluster, supported by the Research Computing Group at Harvard Medical School.

### Review history
The review history is available as Additional file 7.

### Peer review information

### Authors' contributions
KB, MB, and GK designed the study, contributed to the interpretation of the results, wrote the manuscript, and approved the final manuscript. KB developed the software and performed the data analysis. KB and GK developed the theory. The authors read and approved the final manuscript.

### Authors' information
Twitter handles: @KarelBrinda (Karel Břinda); @Baym (Michael Baym); @GregoryKucherov (Gregory Kucherov).

### Funding
KB and MB were partially supported by the David and Lucile Packard Foundation and NIGMS of the National Institutes of Health under award number R35GM133700. GK was partially funded by RFBR, project 20-07-00652, and joint RFBR and JSPS project 20-51-50007.

### Availability of data and materials
Data:
All data generated or analyzed during this study are included in this published article and its supplementary information files. The simplitigs of the human genome (HG38, for $k$ = 10, 11, …, 32) and the obtained GenBank pan-genomes (for $k$ = 19, 23, 27, 31) are provided on Zenodo ([65, 99], respectively). The code used for the analyses is provided on GitHub [100].
Software:
ProphAsm is open source, licensed under the MIT License. The program was developed in C++, and its source code is available from GitHub [44]. ProphAsm binaries for Linux and OS X are distributed through BioConda [98] (https://bioconda.github.io/recipes/prophasm/README.html). The source code of the version used in this paper was deposited in Zenodo [45].

### Ethics approval and consent to participate
Not applicable.

### Consent for publication
Not applicable.

### Competing interests
The authors declare that they have no competing interests.

### Author details
[1]Department of Biomedical Informatics and Laboratory of Systems Pharmacology, Harvard Medical School, Boston, USA and Broad Institute of MIT and Harvard, Cambridge, USA. [2]Center for Communicable Disease Dynamics, Department of Epidemiology, Harvard T.H. Chan School of Public Health, Boston, USA. [3]CNRS/LIGM Univ Gustave Eiffel, Marne-la-Vallée, France. [4]Skolkovo Institute of Science and Technology, Moscow, Russia.

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

## 
