## [**Additional file 7.** Review history. · Genome Biology]

Review History

First round of review

Reviewer 1

Were you able to assess all statistics in the manuscript, including the appropriateness of statistical tests used? No.

Were you able to directly test the methods? No.

Comments to author:

The paper proposes simplitigs, which are an extension of unitigs in de Bruijn graphs representing k-mers in DNA sequences.

The proposed representation of de Bruijn graphs reduces the space requirement of the graphs. Experimental results show that using simplitigs can reduce memory consumption and also running time. Therefore the proposed scheme will be practical and useful.

However there are some questions.

1. Why do not you compare your scheme with succinct de Bruijn graphs [11, 59]?
2. The reviewer agrees simplitigs use less space than unitigs, but does not understand why the algorithm using simplitigs is much faster than that using unitigs. Are the running times proportional to the memory usage?

Reviewer 2

Were you able to assess all statistics in the manuscript, including the appropriateness of statistical tests used? No.

Were you able to directly test the methods? Yes.

Comments to author:

The authors present an approach to generate a compacted k-mer set representation of a de Bruijn graph. The concept is similar to the disjointig concept used by Flye for assembly to construct repeat graphs (as mentioned by the authors) as well as the best overlap graph approach used by Celera Assembler on overlap graphs. I was able to download the run the software and the method was clearly presented.

- The background and description both state that simplitigs are a generalization of unitigs and focus on assembly. I found this confusing because unitigs are typically defined as occurring in all possible genomic reconstructions. However, simplitigs as defined by the authors, are not guaranteed to occur in all such reconstructions. There has been other prior work on unitig generalization which maintains the above property (e.g. omnitigs) but I think the introduction and discussion should be updated to make clear the goal here is to preserve only the k-mers in the graph as compactly as possible, without caring about the "true" genome reconstruction.

- The background would be better re-written to focus on other similar compression approaches. The most similar one (<https://www.biorxiv.org/content/10.1101/2020.01.07.896928v1.abstract>) is already cited but only mentioned at the very end. It would be useful to both mention it earlier but also include its results in the evaluation. It may be less CPU and memory efficient, as the authors say, due to its reliance on unitigs, but that is never actually demonstrated on real data. The comparison of NS and CL between the methods

would also be interesting. There have been some other SPSS publications (e.g. Blight Marchet et al. 2019), including some which attempted to include count information (e.g. Marchet et al. 2020) that should be cited as well.

- While not directly comparable to the simplitig approach, graph-based aligner attempt to gain similar efficiency of compacting multiple assemblies down to a more efficient structure as in the pan-genome example. From a practical use perspective, it would be interesting to include a comparison to a graph-based aligner (such as vg) in this section so readers know how this indexing with a traditional aligner (e.g. BWA) compares to graph-based aligners.

- Algorithm 1 seems to have a typo in the `compute_maximal_simplitig_from_kmer` should be `simplitig = seeding_kmer` not `simplitig = initial_kmer`

General answer

We would like to thank both reviewers for their time, effort and insightful comments. Their feedback inspired substantial additional work that we think has greatly improved the manuscript. Major updates compared to the previous submission are the following:

- Additional experiments done on comparing compression capabilities of simplitigs to other approaches, presented in a new section “Application of simplitigs for de Bruijn graph storage”
- Updated Background, Discussion, and Conclusions
- Improved organization and clarity throughout the whole paper

Editor

Dear Dr. Břinda,

Thank you very much for submitting your manuscript entitled 'Simplitigs as an efficient and scalable representation of de Bruijn graphs' to Genome Biology, and please accept my apologies for the delay in replying to you about it. It has now been seen by two referees and their comments are accessible below.

As you will see from the reports, both referees are favorable and find the work of potential interest, but they raise important issues that we must ask you to address, in the form of a revised manuscript, before we reach a final decision on publication. In particular, Reviewer 2 suggests additional discussion of similar approaches, and both reviewers suggest some additional comparisons. Please ensure that these and all other issues raised by the referees are addressed in full.

We are considering your manuscript as a 'Method' article. This is our format for publishing articles that describe a methodological innovation that is a significant advance over published methods and likely to be of broad utility, but that do not provide significant biological insights. When revising the manuscript, please ensure the manuscript conforms to our style for Methods articles (see <https://genomebiology.biomedcentral.com/submission-guidelines/preparing-your-manuscript/method>); specifically, the abstract should be under 100 words. Please note that if we decide to publish your manuscript we will require that the source code is made publicly available under an open source license compliant with Open Source Initiative, with the license clearly stated in the manuscript. The source code should be deposited in a public repository, such as for instance github, with the accession links included in the manuscript. We also ask that the version of source code used in the manuscript is deposited in a DOI-assigning repository, such as zenodo, with the link also included. All this information should be listed in a separate Availability of Data and Materials section of the manuscript.

We have shortened the abstract to 100 words. All the source code is available on Github under the MIT license (<https://github.com/prophyle/prophasm>, <https://github.com/karel-brinda/simplitigs-supplementary>), and the ProphAsm source code is also available on Zenodo (<https://doi.org/10.5281/zenodo.3887035>). The accession codes and urls are listed in the section “Availability of data and materials”.

Reviewer 1

The paper proposes simplitigs, which are an extension of unitigs in de Bruijn graphs representing k -mers in DNA sequences. The proposed representation of de Bruijn graphs reduces the space requirement of the graphs. Experimental results show that using simplitigs can reduce memory consumption and also running time. Therefore the proposed scheme will be practical and useful. However there are some questions.

1. Why do not you compare your scheme with succinct de Bruijn graphs [11, 59]?

We have added a section "Application of simplitigs for de Bruijn graph storage", where we compared the size of compressed simplitigs to the size of the compressed BOSS representation (as implemented in Themisto, as we were unable to compile the original implementation from <https://academic.oup.com/bioinformatics/article/33/20/3181/2995815> on our computers). In this section we clarify that succinct de Bruijn graphs are a specialized membership data structure rather than a representation of de Bruijn graphs in our sense; therefore, these two representations are not directly comparable. However, the performance can be compared for specific applications, as we did in this experiment.

2. The reviewer agrees simplitigs use less space than unitigs, but does not understand why the algorithm using simplitigs is much faster than using unitigs. Are the running times proportional to the memory usage?

We have clarified in the text that the observed improvement in running times likely arose from a combination of factors, including the lower number of sequences (NS), the reduction of high-frequency spurious k -mers on sequence borders in the composite sequence, and also the lower memory footprint (as noted by the reviewer). Overall, simplitigs improve many parameters related to these factors, but the final improvement also depends on the hardware (e.g., the sizes of caches). More detailed analysis will appear in our future work, which will focus on the specific end-to-end protocols and will provide a tool specifically optimized for k -mer matching using simplitigs.

Reviewer 2

The authors present an approach to generate a compacted k -mer set representation of a de Bruijn graph. The concept is similar to the disjointig concept used by Flye for assembly to construct repeat graphs (as mentioned by the authors) as well as the best overlap graph approach used by Celera Assembler on overlap graphs. I was able to download the run the software and the method was clearly presented.

- The background and description both state that simplitigs are a generalization of unitigs and focus on assembly. I found this confusing because unitigs are typically defined as occurring in all possible genomic reconstructions. However, simplitigs as defined by the authors, are not guaranteed to occur in all such reconstructions. There has been other prior work on unitig generalization which maintains the above property (e.g. omnitigs) but I think the introduction and discussion should be updated to make clear the goal here is to preserve only the k -mers in the graph as compactly as possible, without caring about the "true" genome reconstruction.

We have improved Background, Discussion, and Conclusion to increase clarity on this point. We tried to avoid possible confusions with assembly applications and assembly-centric unitig generalizations such as omnitigs.

- The background would be better re-written to focus on other similar compression approaches. The most similar one (<https://www.biorxiv.org/content/10.1101/2020.01.07.896928v1.abstract>) is already cited but only mentioned at the very end. It would be useful to both mention it earlier but also include its results in the evaluation. It may be less CPU and memory efficient, as the authors say, due to its reliance on unitigs, but that is never actually demonstrated on real data. The comparison of NS and CL between the methods would also be interesting.

In the updated Background, we highlighted the central idea of simplitigs being a lightweight representation carrying the same k -mers. As the associated compressibility is one of the useful properties, we added a section comparing simplitig-based compression to other compression approaches. We also updated Discussion to explain the relation between simplitigs and SPSS (they are the same thing), heuristic for their computation (a greedy heuristic in both cases) as well as the difference in required resources (i.e., ProphAsm vs. BCALM in Figure 3 as the SPSS pipeline uses BCALM).

There have been some other SPSS publications (e.g. Blight Marchet et al. 2019), including some which attempted to include count information (e.g. Marchet et al. 2020) that should be cited as well.

We have added the missing citations (in Background and Discussion).

- While not directly comparable to the simplitig approach, graph-based aligner attempt to gain similar efficiency of compacting multiple assemblies down to a more efficient structure as in the pan-genome example. From a practical use perspective, it would be interesting to include a comparison to a graph-based aligner (such as vg) in this section so readers know how this indexing with a traditional aligner (e.g. BWA) compares to graph-based aligners.

We agree that a comprehensive comparison of simplitig-based approaches to graph-mapper-based ones in the context of bacterial pan-genomics would be of high value to the community. However, the current state of graph-based methods didn't allow us to perform such a comparison on our data. Even though graph references have been recently adopted in human genomics (<https://www.nature.com/articles/nbt.4227>), bacteria are more challenging due their high variability, and reference graph construction for individual bacterial species has not yet been sufficiently solved. For instance, while the VG mapper could, in principle, work with bacterial pan-genomes, the associated methods have not yet been fully developed, as discussed on Github (<https://github.com/vgteam/vg/issues/1589>). We foresee the Pandora tool (<https://ora.ox.ac.uk/objects/uuid:a8027db9-8d8c-4015-aab6-1371217ccb5c>) to be the most appropriate software for such a comparison, but this tool has not yet been published and the preprint appeared just several days ago. We have added a note to this regard in Discussion.

- Algorithm 1 seems to have a typo in the `compute_maximal_simplitig_from_kmer` should be `simplitig = seeding_kmer` not `simplitig = initial_kmer`

We have corrected the line.